# “Difficult to Sedate”: Successful Implementation of a Benzodiazepine-Sparing Analgosedation-Protocol in Mechanically Ventilated Children

**DOI:** 10.3390/children8050348

**Published:** 2021-04-28

**Authors:** Nataly Shildt, Chani Traube, Mary Dealmeida, Ishaan Dave, Scott Gillespie, Whitney Moore, Lillian D. Long, Pradip P. Kamat

**Affiliations:** 1Department of Pediatrics, Emory University School of Medicine, Atlanta, GA 30322, USA; ndshildt@gmail.com (N.S.); lihinie.deAlmeida@choa.org (M.D.); 2Division of Pediatric Critical Care Medicine, Children’s Healthcare of Atlanta, Atlanta, GA 30322, USA; 3Department of Pediatrics, Weill Cornell Medical College, New York, NY 10065, USA; chr9008@med.cornell.edu; 4Pediatric Biostatistics Core, Department of Pediatrics, Emory University School of Medicine, Atlanta, GA 30322, USA; ishaan.dave@emory.edu (I.D.); scott.gillespie@emory.edu (S.G.); 5Children’s Healthcare of Atlanta, Atlanta, GA 30322, USA; Whitney.moore@choa.org; 6Georgia Institute of Technology, Atlanta, GA 30332, USA; lillian.long83@gmail.com

**Keywords:** analgosedation, benzodiazepines, delirium, opiates, sedation, withdrawal, protocol

## Abstract

We sought to evaluate the success rate of a benzodiazepine-sparing analgosedation protocol (ASP) in mechanically ventilated children and determine the effect of compliance with ASP on in-hospital outcome measures. In this single center study from a quaternary pediatric intensive care unit, our objective was to evaluate the ASP protocol, which included opiate and dexmedetomidine infusions and was used as first-line sedation for all intubated patients. In this study we included 424 patients. Sixty-nine percent (*n* = 293) were successfully sedated with the ASP. Thirty-one percent (*n* = 131) deviated from the ASP and received benzodiazepine infusions. Children sedated with the ASP had decrease in opiate withdrawal (OR 0.16, 0.08–0.32), decreased duration of mechanical ventilation (adjusted mean duration 1.81 vs. 3.39 days, *p* = 0.018), and decreased PICU length of stay (adjusted mean 3.15 vs. 4.7 days, *p* = 0.011), when compared to the cohort of children who received continuous benzodiazepine infusions. Using ASP, we report that 69% of mechanically ventilated children were successfully managed with no requirement for continuous benzodiazepine infusions. The 69% who were successfully managed with ASP included infants, severely ill patients, and children with chromosomal disorders and developmental disabilities. Use of ASP was associated with decreased need for methadone use, decreased duration of mechanical ventilation, and decreased ICU and hospital length of stay.

## 1. Introduction

Recent studies have demonstrated a consistent and robust association between exposure to benzodiazepines and delirium development in pediatric critical illness [1,2,3,4,5]. As pediatric delirium is related to poor outcomes (including increased duration of mechanical ventilation, increased ICU and hospital length of stay, and even excess mortality), there is a compelling need to explore alternatives to benzodiazepine-based sedation in mechanically ventilated children [6,7,8].

However, benzodiazepines (specifically, midazolam infusions) have been the mainstay of pediatric sedation protocols for many decades [9,10]. As of 2013, most pediatric critical care units were still using midazolam as first-line sedation in mechanically ventilated children [11,12,13]. Clinicians often prefer midazolam infusions because of perceived benefits and are especially wary of benzodiazepine-free sedation in children who are assumed to be “difficult to sedate” (i.e., young children and those with chromosomal disorders or other developmental disabilities) [10,14]. The perceived benefits of midazolam infusion from previous studies include minimal physiological alteration in the critically ill patient, short duration of action without accumulation or active metabolites, flexibility in route of administration (can be given enterally, intranasally, rectally) if intravenous access is interrupted. Additionally, midazolam infusions allowed for the provision of enteral nutrition in critically ill patients [12,13]. However, limited evidence exists as to which critically ill children will succeed with a benzodiazepine-sparing analgosedation (ASP) protocol during mechanical ventilation.

The primary objective of this study was to describe the rate of children who were successfully sedated using a benzodiazepine-infusion-free ASP and report the association between use of the ASP and in-hospital outcomes (mortality, unplanned extubation, duration of mechanical ventilation, length of stay, and need for opiate withdrawal treatment with methadone). ASP-cohort was defined as patients whose SBS was maintained at goal using the ASP for the entire duration of invasive mechanical ventilation. Opioid withdrawal was defined as the need for initiation of methadone treatment during the PICU stay. Successful sedation in a patient was defined as the patient who maintained SBS using ASP for their duration of mechanical ventilation without the need for benzodiazepines infusions or other sedatives.

A secondary objective was to describe the cohort of children who required deviation from the ASP, with the initiation of a continuous benzodiazepine infusion. Benzodiazepine-cohort was defined as patients who received a benzodiazepine infusion for >24 h to maintain the desired SBS.

We hypothesized that the ASP would be successfully used in most of our mechanically ventilated patients and that use of the ASP would be associated with decreased duration of mechanical ventilation, decreased ICU length of stay (LOS), and no increase in unplanned extubations or mortality rates. We were agnostic as to the effects of the ASP on the development of opiate withdrawal, as we thought that opiate use might increase with the ASP (to replace some of the sedative effects of benzodiazepines), but this could be balanced by a decreased duration of mechanical ventilation. We further hypothesized that children less than three years of age, those with the highest severity of illness, and those with underlying neurologic disorders would be at higher risk of deviation from the ASP.

## 2. Materials and Methods

### 2.1. Patient Population

This single-site retrospective cohort study took place in an academic quaternary care level I trauma medical-surgical pediatric intensive care unit (PICU) with more than 3500 admissions per year. All patients (0–18 years of age) requiring invasive mechanical ventilation and admitted to the PICU between 1 March 2018 and 31 March 2019 were eligible for inclusion. Patients admitted with a diagnosis of status epilepticus, status asthmaticus requiring isoflurane, traumatic brain injury, cardiac arrest, and those who required infusions of a neuromuscular blocking agent were excluded from this study. From our previous experience, the above patients frequently require benzodiazepine infusions for seizure control, which could obscure our data and create confusion about our results. These patients would receive a benzodiazepine not primarily for sedation but for their neurological status.

Additionally, postoperative patients with critical airways (such as fresh tracheostomy or laryngotracheal reconstruction) were excluded. Most of these patients receive a neuromuscular blocking agent in first 4–5 days prior to the scheduled tracheostomy change due to airway safety. We use benzodiazepine infusions in these patients prior to using neuromuscular blocking agents. The Children’s Healthcare of Atlanta Institutional Review Board approved this minimal-risk study with a waiver of the requirement for informed consent (IRB # STUDY00000118).

### 2.2. Analgosedation Protocol

In September of 2017, we developed a nurse-driven ASP (Appendix A) with the basic aims of optimizing pain management and limiting exposure to benzodiazepine infusions. Analgesics used in our PICU include morphine, fentanyl, and hydromorphone. Sedatives used include benzodiazepines such as midazolam, and lorazepam. We also rarely use a dissociative agent like ketamine and a barbiturate such as pentobarbital.

For patients expected to extubate within 48 h (such as post-procedural, ingestions and overdose), an infusion of dexmedetomidine with an intermittent, as needed intravenous opiate (morphine) was used. Morphine was selected for its longer duration of action as an analgesic agent. For patients less than 20 kg, 0.05 mg/kg intravenous morphine bolus was used as needed every 3 h. For patients > 20 kg, 2–4 mg intravenous morphine boluses were used as needed every 3 h.

For those patients who were expected to remain intubated > 48 h, an opiate (fentanyl) infusion was initiated as first-line therapy. For patients less than 20 kg, an initial loading dose of fentanyl of 1 mcg/kg was administered and repeated every 5 min three times followed by a fentanyl infusion at 1 mcg/kg/h. The fentanyl infusion was subsequently titrated upwards by 0.5 mcg/kg if >3 additional bolus doses were required in one hour to maintain the patient’s SBS till a maximum of 5 mcg/kg of fentanyl infusion was reached. For patients > 20 kg, an initial loading dose of 25–50 mcg was administered every 5 min three times followed by an initial infusion of 25–50 mcg/h. The fentanyl infusion was subsequently titrated upwards by 25 mcg/h if >3 additional bolus doses were required in one hour to maintain patient’s SBS till a maximum of 200 mcg/h.

Dexmedetomidine infusion was added if there was an inability to maintain the State Behavioral Scale (SBS) at the desired level [15]. If a patient developed tachyphylaxis to the opiate agent being used a common occurrence with opiates especially fentanyl, a decision to switch to a different opiate agent (morphine or hydromorphone) was made at the discretion of the attending and pharmacist [16,17]. For this study we considered tachyphylaxis as the inability to achieve target SBS with the opiate in use despite increasing the infusion rate or reaching the maximal infusion rate. For patients < 20 kg, morphine was administered as an initial loading dose of 0.1 mg/kg every 10 min three times followed by an initial infusion of 0.1 mg/kg/h, which was subsequently titrated upwards by 0.05 mg/kg/h if >3 additional bolus doses were required in one hour to maintain patient’s SBS till a maximum of 0.3 mg/kg/h. For patient’s > 20 kg, an initial loading dose of 2 mg morphine was administered and repeated every 10 min three times followed by an infusion of 2 mg/h. The infusion was subsequently titrated upwards by 0.5 mg/hour if >3 additional bolus doses were required in one hour to maintain patient’s SBS to a maximum of 5mg/hour. For patients < 20kg, hydromorphone was administered as an initial loading dose of 0.01 mg/kg every 10 min three times followed by an infusion of 0.01 mg/kg/h. The hydromorphone infusion was subsequently titrated upwards by 0.005 mg/kg/h if >3 additional bolus doses were required in one hour to maintain patient’s SBS to a maximum of 0.1 mg/kg/h. For patients > 20 kg, an initial loading dose of hydromorphone of 0.2 mg was administered every ten minutes three times followed by an infusion of 0.2 mg/h. The hydromorphone infusion was subsequently titrated upwards by 0.2 mg/h to maintain patient’s SBS to a maximum of 2 mg/h.

The ASP did not include benzodiazepine infusions as part of its tiered therapies, but it did allow as-needed intermittent benzodiazepine doses if the patient’s SBS exceeded target SBS despite optimizing opiate and dexmedetomidine therapies. We used midazolam or lorazepam intermittently as follows:

For Patients < 20 kg under 6 months of age: Midazolam 0.05 mg–0.1 mg/Kg IV every 3 h as needed. For patients > 6 months: Lorazepam 0.05 mg/Kg–0.1 mg/Kg every 3 h.

For Patients > 20 kg: Lorazepam 2–4 mg IV every 3 h as needed.

All patients on the protocol were ordered to receive a daily sedation holiday (holding of continuous infusions. We perform the daily sedation interruption on every patient on our morning PICU multidisciplinary rounds unless otherwise contraindicated. The nurse allows this sedation drug interruption as long as the patient’s SBS is maintained. If the patient’s SBS is unable to be maintained the nurse will restart the medication at half the previous infusion rate. If after daily sedation interruption, the patient was not agitated or in pain, then an assessment for extubation readiness was performed.

If the ICU physician was not able to achieve the sedation goal with the ASP, s/he was permitted to go “off protocol” and initiate a continuous benzodiazepine infusion. Our pharmacist initiated opiate withdrawal protocol with methadone (intravenously for first 24 h and then followed enteral methadone administration) is based on calculations of total opiate used during mechanical ventilation [18,19,20]. Opiate withdrawal protocol with methadone was started immediately after extubation when ASP was stopped or very rarely the night prior to extubation.

After extensive training of the staff (nurses, respiratory therapists, fellows, and critical care medicine attendings), the ASP was implemented in our PICU. As a part of the protocol, we used the validated State Behavioral Scale (SBS), and the Withdrawal Assessment Tool (WAT) score [15,21]. At the time of implementing our ASP, staff training for delirium recognition using the Cornell Assessment of Pediatric Delirium (CAPD) was not complete; therefore, most patients were not screened for delirium over the course of this study [22]. The severity of illness was defined using the Pediatric Risk of Mortality II (PRISM) scores, with a higher score connoting more severe illness [23].

### 2.3. Data Collection

Demographics included patient’s age, gender, admitting diagnosis, and PRISM score. Clinical data included sedative and analgesic medications, duration of mechanical ventilation, duration of PICU and hospital length of stay, unplanned extubations, and mortality.

### 2.4. Statistical Analysis

Data were summarized by sedation cohorts (i.e., ASP cohort vs. benzodiazepine cohort) using descriptive statistics, including counts and percentages for categorical variables and medians and interquartile ranges for continuous variables, as appropriate. When modeling dichotomous clinical outcomes (e.g., methadone requirement or unplanned extubation) with sedation cohort as the exposure, binary logistic regression was used to compute unadjusted and adjusted associations (adjusted for age at admission and PRISM score). Dichotomous results are presented as odds ratios with 95% confidence intervals and *p*-values. For continuous clinical outcomes, such as length of stay, general linear models were used. For all continuous models, data were transformed by natural log prior to analysis to meet the assumption of error residual normality. Results were back-transformed via exponentiation for interpretation purposes and presented as least squares means (LS-Means) with 95% confidence intervals and *p*-values. When computing associations between patient characteristics as the exposures and sedation cohort as the outcomes, binary logistic regression was similarly employed. All statistical analyses were performed using SAS v. 9.4 (SAS Institute, Cary, NC, USA), and significance was assessed at the 0.05 level.

## 3. Results

### 3.1. Study Cohort

During the study period, a total of 424 patient encounters met inclusion criteria. Fifty-eight percent of the study population was male. The median patient age was 4.1 years. The median PRISM score was 9 (IQR 6–12). The three most common reasons for admission were acute respiratory failure, neurological disorders, and trauma. Seventy-three children (17.2%) had pre-existing developmental disabilities. An additional 21 children (4.9%) had chromosomal disorders. Forty-eight children (11.3%) had a history of premature birth. Only five children (1.1%) had a diagnosis of autism. Four (0.94%) children required continuous veno-venous hemofiltration (CVVH), and five (1.1%) required high-frequency oscillatory ventilation (HFOV) (Table 1).

### 3.2. Medications Used in the Benzodiazepine Cohort

Medications used in the benzodiazepine cohort and their start date from initiation of mechanical ventilation are showed in Table 2. Pentobarbital was used as required and not as an infusion. Patients who required benzodiazepines do so within 1.5 days of initiation of mechanical ventilation.

### 3.3. ASP Implementation (Practicality of a Benzodiazepine-Sparing Approach)

Sixty-nine percent of children (293/424) were successfully sedated using the ASP. Thirty-one percent of children (131/424) received continuous benzodiazepine infusions for >24 h. Neither patient age, sex, nor admission diagnosis associated with need for benzodiazepines. In bivariate analysis, the median PRISM score was slightly higher in the group that required initiation of benzodiazepines (9 vs. 8, *p* = 0.005). Patient demographics and clinical characteristics for the two cohorts are shown in Table 1.

Of note, children with developmental disabilities were no more likely to require initiation of benzodiazepines than children with typical development. For example, in children with developmental delay, the unadjusted odds ratio for successful completion of the ASP was 1.45 (0.81–2.58, *p* = 0.207). The same was true for children with a history of prematurity (OR = 1.23 for successful protocol completion compared to children without prematurity [95% CI 0.63–2.41, *p* = 0.544]). Additionally, neither younger age nor increased severity of illness associated with decreased success for ASP protocol completion. The ASP was successfully used in all three patients on CVVH and 2/5 patients on HFOV.

Receiver operating characteristic (ROC) curves (data not shown) demonstrated that age, severity of illness, developmental delay, chromosomal disorder, autism, and prematurity did not predict the need to deviate from the analgosedation protocol and initiate benzodiazepine infusions. The area under the curve (AUC) for above ROC was less than 0.5.

### 3.4. Association between ASP Use and Clinical Outcomes

Fifteen out of 278 patients (5%) treated with the ASP required methadone for opiate withdrawal, as compared to 33/98 (25%) who were started on benzodiazepine infusions, indicating protective odds of a methadone requirement for the ASP cohort relative to the benzodiazepine infusion cohort (OR = 0.16, 95%: 0.08–0.32, *p* < 0.001). Mortality rates were no different between groups (8% in ASP-cohort vs. 12% in benzodiazepine-cohort; OR = 0.61, 95% CI: 0.29–1.25, *p* = 0.127). There was one unplanned extubation in each cohort (Figure 1). Duration of mechanical ventilation was shorter (LS-Mean: 1.81 vs. 3.39 days, *p* = 0.018), as was PICU length of stay (LS-Mean: 3.15 vs. 4.70 days, *p* = 0.011) and hospital length of stay (LS-mean 8.28 vs. 11.58, *p* = 0.04) in the ASP group (Table 3).

Legend: This forest plot compares odds ratios for methadone requirement, unplanned extubations, and mortality between the analgosedation cohort and the benzodiazepine cohort.

All outcomes were transformed prior to regression modeling by a natural log, to meet the assumption of error residual normality. These results have been back transformed via exponentiation for interpretation purposes.

## 4. Discussion

Children admitted to the PICU frequently require analgesia and sedation in order to tolerate invasive mechanical ventilation [24]. However, a large body of pediatric literature describes morbidity attributable to prolonged benzodiazepine use [7,11]. Despite this, before September 2017, our PICU used a midazolam infusion as first-line sedation in all mechanically ventilated children. However, in this study, we describe the successful implementation of a benzodiazepine-sparing protocol in ~70% of our intubated patients. This suggests that the use of benzodiazepines in PICUs can be dramatically decreased in children on invasive mechanical ventilation.

Contrary to popular opinion, specific patient populations were not more difficult to sedate. Before implementing the ASP in our unit, practitioners were concerned about the “difficult to sedate” patients [14]. Conventional wisdom held that specific patient groups (i.e., infants, children with a developmental disability or history of prematurity, children with severe underlying illness) would prove difficult to control with an analgosedation approach and would routinely require initiation of a continuous benzodiazepine infusion to treat breakthrough agitation. Rather than exclude those children from the ASP a priori, we decided to utilize the ASP in all mechanically ventilated children as first-line sedation and allow for deviation from protocol when necessary. In contrast to our initial hypothesis, these children were successfully managed using the ASP. We were able to maintain appropriate sedation levels, without excess agitation, using an analgosedation approach.

Previous reports have shown that patients with developmental delay and younger children (<5 years of age) are at the highest risk for the development of ICU delirium [5,7]. Thus, the high rate of successful sedation without the use of benzodiazepine infusions in this cohort is opportune. Perhaps what had been previously described as “difficult to sedate” in these children was, rather, unrecognized delirium. In the past, by mistakenly treating the agitated symptoms of delirium with higher doses of benzodiazepines, it is possible that practitioners were unintentionally worsening the underlying delirium and potentiating the agitation [25]. This may have led to an inability to wean the ventilator and increased duration of mechanical ventilation and PICU LOS. We presume that by minimizing benzodiazepine exposure in this study cohort, we decreased delirium risk. Unfortunately, we did not assess delirium rates in this study as we had not started delirium screening in our PICU at the time of data collection. That is a major study limitation and should be an area for focus in future studies.

### 4.1. Patients with Need for Augmented Sedation

Notably, ~30% of children were deviated from the protocol and started on continuous benzodiazepine infusions. The majority of benzodiazepine infusions were initiated during morning PICU rounds, with a multidisciplinary team including the PICU pharmacist, wherein a needs assessment to deepen sedation using benzodiazepines was made. Additionally, nurses underwent an extensive educational program that discouraged the use of benzodiazepine infusions without discussion with an advanced practice nurse, fellow, or attending provider. Hence, we speculate that these protocol ‘deviations’ reflected actual patient-need for augmented sedation. However, it is also possible that it may reflect practitioner comfort with a more traditional midazolam-based sedation algorithm. It is difficult to change the culture and conventional wisdom, but emerging pediatric literature suggests strong benefits associated with a benzodiazepine-sparing analgosedation approach. With further staff education, and ongoing demonstrating of the success of an ASP in most patients, we may be able to further decrease benzodiazepine-exposure in mechanically ventilated children.

Although outside the scope of this study’s objectives, we reviewed subsequent medication exposures in the 131 children who were started on benzodiazepine infusions. The majority of these children did not achieve goal sedation with midazolam; they also required boluses or infusions of additional medications such as ketamine, and pentobarbital in order to achieve target SBS. This suggests that these children were not “benzodiazepine-deficient” but rather represent a cohort refractory to the general sedation approach that will work for most patients. Future research is needed to help define the cohort of children who fail an analgosedation approach and identify best practices for identifying and managing this group so as to optimize outcomes.

### 4.2. Association with Patient Outcomes

It is imperative not to over-conclude based on this retrospective descriptive study. We do not assume a causal relationship between the successful completion of the ASP and the outcomes described. It is entirely possible that the children who required deviation from the protocol and initiation of benzodiazepine infusions were different in substantial ways that were not captured in the PRISM score, demographics, or patient characteristics. At most, we can conclude that the benzodiazepine-sparing protocol was possibly beneficial (with decreased opiate withdrawal, LOS, and duration of MV), and safe (consistent with the existing literature, there was no increase in serious adverse events such as unplanned extubations).

We not only decreased the exposure to benzodiazepines in these high-risk children but also reduced opiate withdrawal rates. This is physiologically plausible, as benzodiazepine use may exacerbate opiate tolerance, leading to an increase in opiate exposure [26]. In addition, benzodiazepines are independently associated with delirium, and delirium is known to prolong the duration of mechanical ventilation [3,5,7]. With a shorter time to extubation, total opiate exposure can be substantially lessened.

### 4.3. Study Limitations

Our study has notable limitations. It is a single-center retrospective cohort with a relatively small sample size, which describes an association with no causal inference. Further research is necessary to establish the generalizability of our findings. Another limitation, as mentioned previously, is that we did not screen for delirium in our cohort. It is possible that the favorable outcomes we describe in association with the ASP may have resulted from decreased delirium risk, and not the ASP itself. It is also possible that children who deviated from our protocol and required benzodiazepines and other sedative medications were, in essence exhibiting hyperactive delirium that was not recognized. With better delirium recognition and management, they may have completed the ASP. Finally, patients on the ASP were allowed short-term use of benzodiazepines (for example, on-demand single-doses of lorazepam). This exposure may have diluted the effect of the ASP. Future studies should consider comparing benzodiazepine-free sedation to any benzodiazepine exposure. In this study we report on the use of methadone as a surrogate marker for iatrogenic drug withdrawal, it is possible that patients could have iatrogenic withdrawal for other medications and such withdrawal is not captured by the Withdrawal Assessment Tool (WAT) score. Additionally, the ASP protocol uses a mcg/hour infusion dosing for fentanyl in children > 20 kg. The mg/hour fentanyl dosing does not take into account the weight-based clearance in younger children who may be bigger in size. However, we used SBS scores in our study, which may have helped guide the dosing titration of fentanyl to achieve therapeutic plasma concentrations [27,28].

## 5. Conclusions

We report on the successful use of a benzodiazepine-infusion free analgosedation protocol even in groups of children previously described as “difficult to sedate”. The analgosedation protocol was associated with benefits including decreased opiate withdrawal, decreased duration of mechanical ventilation, and decreased PICU length of stay. Further studies are needed to closely investigate the cohort, which deviated from the benzodiazepine-infusion free analgosedation protocol.

## Figures and Tables

**Figure 1 children-08-00348-f001:**
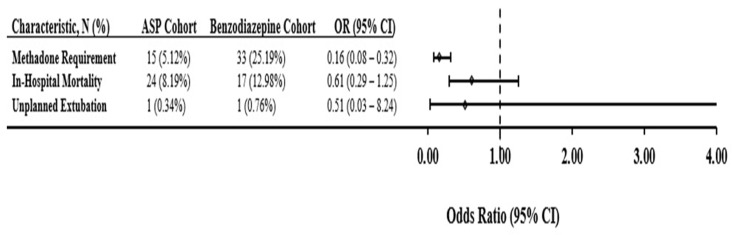
Comparison of odds for adverse events between ASP cohort and Benzodiazepine cohort.

**Table 1 children-08-00348-t001:** Demographic and clinical characteristics of study cohort (*n* = 424).

Variable	Overall (*n* = 424)	ASP Cohort *n* = 293 (69.1%)	Benzodiazepine Cohort *n* = 131 (30.9%)	*p*-Value
Sex, n (%)				
Female	177 (41.75%)	126 (43.00%)	51 (38.93%)	0.432
Male	247 (58.25%)	167 (57.00%)	80 (61.07%)	
Admission Age, Median [IQR]	4.11 [0.95, 11.42]	4.43 [1.04, 11.60]	3.76 [0.84, 10.30]	0.359
PRISM Score	9 [6, 12]	8 [6, 11]	9 [6, 13]	0.005
Primary Diagnosis				
Respiratory	84 (19.81%)	54 (18.43%)	30 (22.90%)	0.140
Neurologic	83 (19.58%)	61 (20.82%)	22 (16.79%)	
Trauma	62 (14.62%)	39 (13.31%)	23 (17.56%)	
Oncology	35 (8.25%)	23 (7.85%)	12 (9.16%)	
Infectious	34 (8.02%)	29 (9.90%)	5 (3.82%)	
Other/Unknown	31 (7.31%)	19 (6.48%)	12 (9.16%)	
Surgical	24 (5.66%)	21 (7.17%)	3 (2.29%)	
Sepsis	25 (5.90%)	16 (5.46%)	9 (6.87%)	
Gastrointestinal/Liver	21 (4.95%)	16 (5.46%)	5 (3.82%)	
Heme	11 (2.59%)	5 (1.71%)	6 (4.58%)	
Cardiac	9 (2.12%)	7 (2.39%)	2 (1.53%)	
Renal	5 (1.18%)	3 (1.02%)	2 (1.53%)	
Co-Morbidities:				
Development Delay	73 (17.22%)	55 (18.77%)	18 (13.74%)	0.205
Prematurity	48 (11.32%)	35 (11.95%)	12 (9.92%)	0.544
Chromosomal Disorder	21 (4.95%)	12 (4.10%)	9 (6.87%)	0.224
Autism Spectrum Disorder	5 (1.18%)	4 (1.37%)	1 (0.77%)	0.601
Advanced Technology:				
CVVH	4 (0.94%)	4 (1.37%)	0 (0.00%)	0.179
HFOV	5 (1.18%)	2 (0.68%)	3 (2.29%)	0.157

IQR = Interquartile range. PRISM = Pediatric Risk of Mortality Score. CVVH = Continuous veno-venous hemofiltration. HFOV = High frequency oscillatory ventilation.

**Table 2 children-08-00348-t002:** Shows medications used in the benzodiazepine cohort.

Drug	Continuous (mg/kg/day)	Time to Initiation from MV (days)
Lorazepam, Median, (IQR) (*n* = 57)	0.37 (0.153, 4.60)	1.56 (0.36, 4.50)
Midazolam (*n* = 125)	0.88 (0.42, 1.78)	0.06 (0.02, 0.53)
Ketamine (*n* = 21)	1.46 (0.91, 2.79)	1.35 (0.14, 6.57)
Pentobarbital (*n* = 14)	--	2.58 (1.32, 6.56)

**Table 3 children-08-00348-t003:** Comparison of Outcomes between ASP cohort and Benzodiazepine cohort.

Duration (Days)	Unadjusted Mean (95% CI)	Unadjusted *p*-Value	Adjusted Mean (95%CI)	Adjusted *p*-Value
ICU LOS				
ASP cohort	3.14 (2.71–3.62)	0.010	3.15 (2.72–3.64)	0.011
Benzodiazepine cohort	4.74 (3.82–5.89)		4.70 (3.78–5.84)	
Hospital LOS				
ASP cohort	8.36 (7.03–9.94)	0.047	8.28 (6.96–9.86)	0.040
Benzodiazepine cohort	11.49 (8.87–14.89)		11.58 (8.93–15.01)	
Mechanical Ventilation				
ASP cohort	1.81 (1.42–2.33)	0.018	1.81(1.41–2.31)	0.018
Benzodiazepine cohort	3.42 (2.43–4.83)		3.39 (2.41–4.77)	

## Data Availability

Study did not report any data in a public domain. However, de-identified data is available with the statisticians at Emory University and Children’s Healthcare of Atlanta Biostatistics Core.

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
