# Peer review of "“Difficult to Sedate”: Successful Implementation of a Benzodiazepine-Sparing Analgosedation-Protocol in Mechanically Ventilated Children"

_children, 2021, doi:10.3390/children8050348_

Round 1

Reviewer 1 Report

I applaud the authors for conducting this study.  With the shift away to benzo-based sedation regimens, I feel that reports like these provide more tangible data to provide to clinicians on regimens that could be implemented in their institutions.  I felt that the manuscript was well written and easy to follow.  However, I have some areas of specific concern that need to be addressed. 

  • Clarification of opioids used as part of the protocol: Given that different opioids have shown different degrees of opioid tolerance, I would like more clarification and transparency on the agents utilized. 
  • Definitions of abstinence and tachyphylaxis: I also feel that the authors need to provide more clarity and explanation of definitions utilized for these terms.  As the methods section currently reads, I have trouble discerning how these were defined and thus have difficulty with the external validity of the findings.
  • Details regarding evaluation of sedation; the sedation is referred to as “successful” several times throughout the article, and a more concrete description of this was would beneficial to the reader.

Specific comments:

  1. Abstract:
    1. Line 18-19: Consider rewording sentence to clarify the study was evaluating the ASP protocol.
    2. Line 25: Sentence appears very generalized in regards to results found, would recommend rewording.
  2. Introduction:
    1. 2nd paragraph:
      1. Line 42-43: “because of perceived safety benefits”, would recommend adding more context to this, as that statement alone does not seem to be entirely true
    2. 3rd paragraph:
      1. Line 48: please provide details of “successfully sedated” and what determined this for patients
      2. Line 51: please clarify “abstinence treatment”; was this methadone initiated after opioids had been stopped or being proactively initiated when on an opioid infusion; please describe how was the need for methadone determined; also consider choosing another term for “abstinence” as it’s meaning appears unclear throughout the article
    3. Methods:
      1. Line 69: Why was the data collection period over 1 year in duration?  This strikes me as odd and arbitrary.  Perhaps more explanation would be beneficial as to why these dates were selected. 
      2. Lines 77-89:
        1. Based on my reading it is hard to tell whether there was a specific opioid infusion that was a part of the protocol. Please clarify.  Based previous data, not all opioids are “created equal” and fentanyl has been associated with a higher degree of opioid tolerance compared to morphine/hydromorphone (Anand KJS, et al.  Pediatr Crit Care Med.  2013;14:27-36 and Ibach BW, et al.  J Pediatr Intensive Care.  2017;6:83-90).  In my opinion, this must be spelled out in the methods section and the results section as it could significantly affect the findings of the study.
        2. Line 81-86: Please provide details of opioid selection, opioid dosing, and titration/bolus need parameters.
  • Lines 83-85: Why were the dosing units changed switched to mg/hr for children >20 kg?  This seems to be a very problematic issue in children given that this would not account for weight-based clearance in younger children who may be bigger in size.  More explanation is needed for this.  See a recent editorial that may shed light on why just approaching dosing on just weight alone should NOT be the appropriate solution for handling dosing in “bigger” kids (Lim SY, et al.  J Pediatr Pharmacol Ther  2018;23:223-6.).
  1. Lines 86-91: The definition of tachyphylaxis seems to be underwhelming.  To me, it is really difficult to assess the external validity of the data in the project based on this definition.  As you are likely aware, there have been previous studies that have assessed opioid tolerance (Anand KJS, et al.  Pediatr Crit Care Med.  2013;14:27-36 and Ibach BW, et al., J Pediatr Intensive Care.  2017;6:83-90., and Lim S, et al.  Analysis of fentanyl Pharmacotherapy  2021. DOI:10.1002/phar.2515).  I would recommend to remove this analysis from manuscript unless you are able to describe a more robust/objective definition. 
  2. Line 91: Please provide details on intermittent benzo use, including benzo selection and dosing/frequency.
  3. Lines 92-94: I commend you on the use of the drug holiday.  However, more details are needed on 1) the duration of time each day that this was initiated and 2) how medications were adjusted.
  1. Lines 103-109:
    1. Would consider moving this section (i.e., “Study Definitions”) earlier in the article since some of this info is already previously discussed
    2. Or consider removing entirely and just providing definitions when discussed throughout the article
  • Why do you use the term “abstinence”? This does not seem to be consistent with the literature (i.e., drug withdrawal, iatrogenic drug withdrawal).  Given the ongoing social stigma with drug overdoses and drug abuse, I think that abstinence has a more negative connotation, given that these children in your study had iatrogenic exposure and thus were not taking these agents recreationally.  In addition, I also find the definition of abstinence to be flawed.  Based on previous studies, this definition would ideally be more rigorous if they included the use of a validated tool like the Withdrawal Assessment Tool-1.
  1. Since the patients were likely receiving more dexmedetomidine than pre-protocol initiation, did you evaluate the number of patients initiated on clonidine? Just curious if you saw more initiation of clonidine to prevent dexmedetomidine withdrawal with the non-benzo based regimens.
  1. Results: Overall, I thought that the results section was well written.  However, my main concerns were again the lack of transparency on the selection of opioid infusions that the patients received.  In addition, I find it troubling that the cumulative dosing that these patients received was not provided.  As cumulative dosing and duration of sedatives and opioids has been associated with the development of withdrawal and tolerance, I find it difficult to wrap my head around the global outcomes without interpreting it in the context of dosing.
  2. Discussion:
    1. Practically of a benzo-sparing approach
      1. Line 203-205: would recommend including this finding in the results section or removing discussion of it in this section
    2. Association with patient outcomes
      1. Line 255: “reduced opioid abstinence rates” may be unclear to the reader; recommend rewording to better demonstrate this decreased the need for methadone for treating IWS
    3. Study limitations
      1. Line 271: consider rewording sentence and avoiding term “Watered down” as this may be unclear or appear informal to the reader
    4. Conclusion: Would consider providing a one sentence summary within conclusion for limitations/future directions.
    5. References: I have noted no specific areas of concern.
    6. Tables/Figures
      1. Figure 1: I found Figure 1 to be blurry and hard to read. Perhaps it is just the pre-production/peer review version of the manuscript.
      2. Table 1: line 146: recommend capitalizing “ Continuous” for consistency

Table 2:  I have noted no specific areas of concern.

Author Response

Dear Reviewer # 1:

We thank you for your suggestions to improve our manuscript. Attached our responses to your queries.

Thanks

Pradip Kamat

Reviewer 2 Report

This is an interesting retrospective single centre study evaluating an analgosedation protocol in mechanically ventilated children. The manuscript in is well written in general. I have some questions and suggestions for the authors to consider, to improve the clarity of the paper:

  • The investigators’ primary hypothesis it that ASP would be “successfully used” – success may be measured in different ways. Suggest that the investigators clarify what they mean by success – e.g. % adoption, adherence, compliance? Reduction in total sedation exposure? Vs “feasible to implement”
  • Given that the hypothesis includes risk of deviation from the ASP, suggest to include the rationale for their exclusion criteria
  • As "successful" use of the ASP is the stated primary objective /hypothesis, some details of the ASP implementation should be provided
  • Some questions regarding the ASP itself, why is DXM listed as first line in the < 48h and second line for the > 48h intubated cohort? How is the clinician to distinguish between “tachyphylaxis” to opioids (in which case they are prompted to use another opioid agent), and the “difficult to sedate” patient, in which case they are prompted to escalate down the protocol, and add on another agent? This is a potentially important concept given their hypotheses.
  • Suggest to list the primary and secondary endpoints clearly in the methods

Results:

  • What is meant by 424 “patient encounters”? are these admissions, patients, sedation events?
  • There is duplication in the reporting in text and Table 1, Figure 1. Suggest to reduce the duplication.

Discussion

  • One of the limitations of this study is the lack of explanation and analyses for the “difficult to sedate” cohort. The ROC curves are eluded to here, yet data is not shown. This is important information and aligned with your secondary objective. Suggest that it should be presented. over fig 1 for example, as those results were already explained in text
  • Why is review of medication exposures outside the scope of this study? The total exposure to medication classes, within and beyond the ASP is an important measure of “success”. While ASP does not allow for benzo infusion, it does allow for prn midazolam and long acting lorazepam. The clinician would be interested in the differences/predictors of those who are refractory to ASP. Arguably - this is part of the primary and the stated secondary objective - not just to define success, but reasons for non-success. ie the "difficult to sedate" cohort.
  • The authors are careful not to overstate their conclusions – this is important as in the instance of Methadone use as a surrogate for iatrogenic withdrawal syndrome, this is dependent on how one weans sedation, and not just the exposure, and what the patient may be withdrawing from if exposed to multiple classes of sedatives. This should be acknowledge.
  • The biggest limitation in my opinion is the comparator cohorts – as it they are comparing ASP to those refractory to ASP, which be design then are set up to have different outcomes. This is not a straight comparison between ASP vs benzo cohort, rather it’s ASP vs ASP + benzo’s cohort. It should be acknowledge why the investigators could not compare ASP vs no ASP.
  • Terms “agnostic” and “abstinence” may not be well understood. Suggest using more traditional terms e.g. “iatrogenic opioid withdrawal” in the case of the latter.

Author Response

Dear Reviewer # 2:

We thank you for your suggestions. Attached we have given our responses to your queries.

Thanks

Pradip Kamat

Reviewer 3 Report

Thank you for allowing me to review this manuscript. Thanks to the authors for this study.

The authors conducted a retrospective single-site cohort study to examine the effect of a benzodiazepine-sparing analgosedation-protocol (ASP) in mechanically ventilated children (opiate and dexmedetomidine infusions as first-line sedation) compared to acohort of children who received continuous benzodiazepine infusions. They reported the successful use of the ASP: decreased abstinence, decreased duration of mechanical ventilation, and decreased PICU length of stay.  31% deviated from the ASP and received benzodiazepine infusions. This is one of the few studies that shows an effect on outcome.

I still have a few questions

  • Were patients in the study group less deeply sedated than in the control group?
  • Target range of the State Behavioral Scale was -1 to -2 in the study group. What was the range in the control group?
  • Did the children in the control group also receive additional to benzodiazepines dexmedetomidine?
  • Do opiate doses differ between groups?
  • Were in the control-group patients, who required boluses or infusions of additional medications to achieve target SBS?

Author Response

Dear Reviewer # 3:

Attached are our responses to your queries. We appreciate your suggestions to improve our manuscript.

Thanks

Pradip Kamat

Round 2

Reviewer 1 Report

None

Author Response

none